# Synergistic Interaction of *Rhizobium tropici*, *Rhizophagus irregularis* and *Serendipita indica* in Promoting Snap Bean Growth

Hayet Beltayef [1,2,*], Mouna Mechri [3], Wafa Saidi [1], Taqi Raza [4,*], Rim Hajri [5], Afef Othmani [6], Khedija Bouajila [7], Cristina Cruz [8], Abeer Hashem [9], Elsayed Fathi Abd_Allah [10] and Mongi Melki [1]

1 Research Laboratory: Support for Sustainable Agricultural Productivity in the Northwest Region, Higher School of Agriculture, University of Jendouba, El Kef 7119, Tunisia; saidifoufa@gmail.com (W.S.); melkimongi59@gmail.com (M.M.)
2 Faculty of Sciences of Bizerte, University of Carthage, Tunis 1054, Tunisia
3 National Institute of Field Crops, Boussalem 8170, Tunisia; mounassol.mm@gmail.com (M.M.)
4 The Department of Biosystems Engineering & Soil Science, University of Tennessee, Knoxville, TN 37996, USA
5 Research Laboratory of Agricultural Production Systems and Sustainable Development, College of Agriculture, The University of Carthage, Mograne Zaghouan 1121, Tunisia; rimhajri@yahoo.fr (R.H.)
6 Field Crops Laboratory, LR20-INRAT-02, National Agricultural Research Institute of Tunisia, University of Carthage, Ariana 2049, Tunisia; othmaniafef@yahoo.com (A.O.)
7 Soil Management, Regional Commissary for Agricultural Development of Beja-Northern Tunisia, Ministry of Agriculture, Hydraulic Resources and Fisheries, Beja 9000, Tunisia; bouajilakhedija@yahoo.com (K.B.)
8 cE3c—Centre for Ecology, Evolution and Environmental Changes & CHANGE, Global Change and Sustainability Institute, Faculdade de Ciências, Universidade de Lisboa, Campo Grande, Bloco C2, 1749-016 Lisboa, Portugal; ccruz@fc.ul.pt (C.C.)
9 Botany and Microbiology Department, College of Science, King Saud University, P.O. Box 2460, Riyadh 11451, Saudi Arabia; habeer@ksu.edu.sa (A.H.)
10 Plant Production Department, College of Food and Agricultural Sciences, King Saud University, P.O. Box 2460, Riyadh 11451, Saudi Arabia; eabdallah@ksu.edu.sa (E.F.A.)
* Correspondence: hayet.beltayef@fsb.ucar.tn (H.B.); taqiraza85@gmail.com (T.R.)

**Abstract:** The overuse of chemical pesticides and fertilizers in crop farming has led to a decrease in crop quality and negative impacts on soil and the environment. It is crucial to adopt alternative strategies to maintain soil and environmental quality while enhancing crop growth and yield. To explore this, a study was conducted under greenhouse conditions to investigate the effect of *Rhizobium tropici* CIAT 899 alone, as well as in association with mycorrhizae (*Rhizophagus irregularis*) and endophytic fungus (*Serendipita indica*), on the growth, yield, and nutrient status of snap bean plants. At harvest, the rhizobial strain CIAT 899 demonstrated the highest effectiveness. It significantly increased the number of nodules in both Contender and Garrafal Enana varieties by 6.97% and 14.81%, respectively, compared with the control without inoculation. Furthermore, the results indicated that co-inoculation of *Rhizobium* and symbiotic fungi had positive effects on nitrogen content, phosphorus availability, and overall plant growth. Regardless of the variety, plants inoculated with *R. tropici* CIAT 899 and *Serendipita indica* exhibited the highest values for plant growth parameters. This combination resulted in 168% and 135% increases in root dry biomass, as well as 140% and 225% increases in the number of pods for Contender and Garrafal Enana, respectively, compared with the control at harvest. Additionally, this study highlights the potential benefits of combining *R. tropici* with either *Serendipita indica* or *Rhizophagus irregularis* in terms of nitrogen and phosphorus uptake. These symbiotic microorganisms demonstrated synergistic interactions with snap bean plants, leading to improved mineral nutrition and enhanced growth. Overall, these findings suggest that utilizing these symbiotic microorganisms can effectively enhance the mineral nutrition and growth of snap bean plants.

**Keywords:** mycorrhizae; endophytic fungus; mineral nutrition; plant–microbe interactions

## 1. Introduction

*Phaseolus vulgaris* L. (common bean) is among the most important legumes with great nutritional value. It plays a crucial role in ensuring global food security [1]. Beans are recognized for their exceptional nutritional value, particularly due to their high protein content, which surpasses that of cereal grains by two to three times. In addition, beans possess elevated levels of dry matter, dietary fiber, minerals, starch, and vitamins in comparison to rice crops [2]. Among the bean varieties, snap beans stand out for their abundance of antioxidant activities and phytochemicals. These include a diverse array of flavonoids such as anthocyanins, proanthocyanidins, flavanols, and phenolic acids, all contributing to snap beans' remarkable nutritional profile [3]. Thus, common bean cultivation is recommended to reduce starvation and food insecurity worldwide [4]. To achieve this, efforts have been made to enhance bean cultivation and increase productivity globally [5]. In fact, the adoption of chemical inputs in agriculture after the green revolution aimed to achieve higher agricultural yields. Meanwhile, the excessive use of chemical inputs like fertilizers, herbicides, and pesticides has detrimental effects on soil health, environmental quality, plant growth, nutrient availability, and crop yield [6]. For example, in Tunisia, the total use of pesticides has been steadily increasing, from 909 tons in 1990 to 2175 tons in 2010 and further rising to 3511 tons in 2020 [7]. Globally, the total use of pesticides has reached 3.5 million tons, a 4 percent increase in a year, 11 percent in a decade, and a doubling since 1990 [8]. Excessive chemical use disrupts the ecosystem's balance and functionality, leading to environmental pollution [9]. Residual and unused chemicals become pollutants in the air, water, and soil [10]. Over the past 400 million years, mycorrhizal fungi and rhizobacteria have played a vital role in maintaining soil health and promoting plant growth and productivity [11]. The use of these microorganisms is a significant advancement in modern farming practices [12]. These plant-growth-promoting microorganisms (PGPMs) have been recognized for their potential to enhance crop productivity, nutrient uptake, disease resistance, and overall plant health [13,14]. As it is known that organic matter, specifically the carbon fraction, plays a critical role in preserving soil health [15,16]. The presence of these microorganisms enhances soil structure, increases organic matter levels, and improves nutrient cycling. Consequently, this leads to long-term advantages in terms of soil health and fertility, thereby encouraging the adoption of sustainable agricultural practices [17]. This innovative approach has profound implications for sustainable agriculture and food security [18,19]. Molecular advances over the last decade have led to new insights into the ecology of mycorrhizal fungi, such as the obligatory biotrophism of mycorrhizal fungi and their ability to adapt to terrestrial ecosystems by colonizing, often simultaneously, a wide range of plant species [20]. Mycorrhizae establish a mutualistic symbiosis with plant roots and can colonize around 80% of terrestrial plants [21]. They are known for their ecological importance. Fungi, through their mycelium, stimulate plant growth by expanding the root surface area [22]. Similarly, rhizobacteria play a vital role in enhancing nutrient availability in the rhizosphere, the soil region surrounding plant roots [23,24]. These beneficial microorganisms engage in various mechanisms that improve nutrient access for plants, particularly for nutrients that may be limited or scarce in the soil [25]. In this interaction, symbiotic fungi receive approximately 20–25% of photosynthetic carbohydrates from plants, as fungi are unable to synthesize them on their own [26]. These fungi, acting as plant-growth promoters, enhance the availability and uptake of multiple essential nutrients, particularly carbon, nitrogen, and phosphorus [27]. They also act as bioprotectors, safeguarding plants against various biotic and abiotic stresses. Additionally, the utilization of rhizobia as legume bioinoculants is a technique that enhances nitrogen bioavailability by fixing atmospheric nitrogen $N_2$ [28]. When legumes form a dual association with both rhizobia and symbiotic fungi, the uptake of mineral elements, particularly nitrogen and phosphorus, is enhanced, leading to improved crop yields [29]. However, the performance of these interactions varies depending on the plant species and the specific microbial strains involved. Scientific understanding of these microorganisms is increasing. Their applications are expanding. Challenges remain in commercial production, application techniques,

and adaptation to different crops and environmental conditions [30]. So far, most scientific studies have focused on understanding the effects of microorganisms on crop growth in controlled environments [31]. Furthermore, research has shown that interactions between nitrogen and phosphorus can impact root morphology, physiology, and nodule formation in field-grown soybeans [32]. There is also evidence of genotypic variation in microbial colonization associated with the root architecture of soybean plants [33]. Currently, limited studies have been conducted on the effects of mycorrhizal and rhizobial strains on legume cultivation in Tunisia. These studies have the potential to identify specific strains suitable for legume cultivation in the country [34]. Enhancing symbiotic interactions and selecting efficient plant–microorganism association can lead to increased biomass yield and improved natural soil fertilization, reducing the need for chemical inputs [35]. To promote widespread snap bean cultivation in Tunisia, it is crucial to assess the symbiotic community of the species and gather information on its development and symbiotic status at various growth stages. This study examined the interaction between symbiotic fungi, *Rhizobium*, and the snap bean crop. Different microbial inoculation approaches were used: single, double, and triple inoculation. The goal was to understand how these microorganisms influence each other in a tripartite relationship. Simultaneous inoculations were performed using *R. tropici* CIAT 899 and two fungal strains. The study considered the impact of the tripartite symbiosis. Moreover, the effect on plant nutrition of combining *Rhizobium* and arbuscular mycorrhizal fungi (AMF) for phosphorus and nitrogen was investigated.

## 2. Materials and Methods

### 2.1. Plant, Microbial Materials and Experimental Site

The plant material utilized in this study consisted of two varieties of snap beans: Contender and Garrafal Enana. Contender is an early snap bean variety that has been registered as a fixed variety in Tunisia since 2005. It is known for its productivity and traditional cultivation practices. On the other hand, Garrafal Enana is a hardy and traditional variety characterized by vigorous growth and high seed production, contributing to its overall productivity.

During this research, the following microorganisms were utilized: *Rhizobium tropici* CIAT 899 (C), which was obtained from the Spanish Type Culture Collection CECT 4654, and two strains of fungi, i.e., *Serendipita indica* (*Si*), an endophytic fungus obtained from the culture collection of the Ecology Laboratory at the Department of Plant Biology, Faculty of Science, Lisbon, Portugal, and *Rhizophagus irregularis* (*Ri*), an arbuscular mycorrhizal fungus provided as a commercial inoculum by the European Bank of Glomales (BEG 163).

The current research study took place at the experimental station of the Higher School of Agriculture of Kef, located at a longitude of 8.720434° and a latitude of 36.1208467°. For the experiment, soil samples were collected from a depth of 0–20 cm and mixed with sand in a ratio of 3:1. The pots used for the experiment had a height of 20 cm and a diameter of 25 cm at the top and 19 cm at the base. They were filled with the soil mixture at a rate of 6 kg per pot to prepare for the experiment. Before starting the experiment, the soil samples were analyzed to determine their physicochemical properties, which are listed in Table 1.

**Table 1.** Physicochemical properties of the soil used.

| Parameters | Unit | Value |
|---|---|---|
| Sand | | 29.5 |
| Clay | % | 47.0 |
| Silt | | 24.5 |
| Texture | -- | Clay |
| pH | -- | 8.3 |
| Organic matter | % | 1.73 |
| Total N | | 0.04 |
| P | mg kg$^{-1}$ | 180.41 |
| K | | 241.80 |

### 2.2. Plant Growth

The common bean seeds underwent a thorough washing process, first with tap water and then with sterile distilled water, repeated twice. To ensure surface sterilization, the seeds were immersed in 70% ethanol for 1 min, followed by a ten-minute immersion in a 3% sodium hypochlorite solution to eliminate any potential contamination. The seeds were then rinsed five times with sterile distilled water. Subsequently, the sterilized seeds were sown in pots filled with sterile sand. After ten days, the seedlings were transplanted into pots filled with a soil substrate consisting of a mixture of sand and soil in a 3:1 ratio. During the seedling transplantation, symbiotic fungi strains *Serendipita indica* and *Rhizophagus irregularis* (*Ri* and *Si*) as well as *R. tropici* strain CIAT 899 (C) were inoculated. The research trial was conducted using a split-plot design with three blocks and two factors (variety and treatment). During the study, five treatments were maintained to assess the impact of inoculation on the growth and yield parameters of snap bean varieties.

(1) Non-inoculated plants (control),
(2) Plants inoculated with *R. tropici* CIAT 899 (C),
(3) Plants inoculated with *R. tropici* CIAT 899 and *Rhizophagus irregularis* (C+*Ri*),
(4) Plants inoculated with *R. tropici* CIAT 899 and *Serendipita indica* (C+*Si*),
(5) Plants inoculated with *R. tropici* CIAT 899, *Serendipita indica*, and *Rhizophagus irregularis* (C+*Ri*+*Si*).

#### 2.2.1. Inoculation with Bacterial Strains

To initiate *R. tropici* CIAT 899 inoculation, a 2 mL volume of bacterial inoculum containing approximately $10^9$ cells per mL was carefully applied to the roots of each individual plant. The bacterial suspension had been previously cultured on Yeast Extract Mannitol (YEM) medium [36]. Inoculation with *Serendipita indica* involved the addition, directly to the plant roots, of a 2 mL volume of its pre-prepared liquid culture, cultivated in KM medium [37], which had a spore concentration of $5 \times 10^5$ spores/mL. The combined treatment (C-*Ri*-*Si*) involving *R. tropici*, *Rhizophagus irregularis*, and *Serendipita indica* was administered on the same day. The roots were enriched with 50 g of *Rhizophagus irregularis* inoculum in the substrate using the layering method [38], providing approximately 1000 spores, followed by the addition of 2 mL of *Serendipita indica* liquid culture. Lastly, 2 mL of *R. tropici* was introduced around the roots.

#### 2.2.2. Enumeration of Autochthonous Rhizobia

The quantification of native rhizobia was carried out using the most probable number (MPN) technique, which assumes that nodulation of a host plant originates from a single rhizobium [36]. For this study, a greenhouse experiment was conducted using plastic bags filled with sterile sand as the culture medium. Four bean seeds were sown in each plastic bag and placed in the greenhouse. After one week of setting up the system, the number of plants per plastic bag was reduced to one, and they were aseptically inoculated with soil suspensions ranging in concentration from $5^{-1}$ to $5^{-6}$.

To prepare the soil suspension, 30 g of soil was mixed with 100 mL of distilled water. The mixture was allowed to settle for 15 min, and then dilution series were prepared. Four repetitions per dilution were performed, using 1 mL of each dilution to inoculate a plant. The number of native rhizobial populations was calculated using Brockwell's MPN tables [39].

### 2.3. Studied Parameters

#### 2.3.1. Biomass Yield

At both the flowering stage (25 days after transplanting) and the fruiting stage (60 days of growth), the above-ground and below-ground organs of the plants were meticulously separated, and their biomass was measured individually. Additionally, the number of pods was also recorded.

2.3.2. Nodulation

Upon reaching the flowering and harvesting stages, the plant roots underwent extensive washing procedures to remove any soil or debris. Subsequently, nodules were carefully extracted from the roots to determine their numerical abundance per plant. In total, six samples were collected for each treatment, and their mean values were calculated.

2.3.3. Nitrogen and Phosphorus Contents

To analyze the total nitrogen (N) content in the aerial biomass and roots, the powdered samples were finely ground and subjected to analysis using a Kjeltec analyzer unit (Kjeltec 8400 Autoanalyzer, Foss Tecator AB, Höganäs, Sweden) [40].

For the quantification of phosphorus in leaves and roots, a 0.5 g ground sample, previously placed in an oven at 60 °C overnight, was crushed and calcined at successive temperature steps of 150 °C, 250 °C, 300 °C, and 400 °C, each lasting 30 min. It was then further calcined at 550 °C for 2 h. The resulting ash was moistened with 3 mL of distilled water and 1 mL of concentrated HCl solution, followed by heating to 80 °C. The extract was filtered and then reduced to 50 mL by adding distilled water before the assay determination. The total phosphorus was determined calorimetrically using the Murphy and Riley method, with a spectrophotometer reading at a wavelength of 882 nm [41].

2.3.4. Mycorrhization Rate

To estimate the mycorrhizal colonization frequency, root fragments were washed and stained with trypan blue. Thirty root fragments per sample, each measuring 1 cm in length, were randomly selected at different levels and observed under a microscope [42]. The frequency of mycorrhizal colonization (M%) was calculated using the following formula:

M% = (number of mycorrhizal root fragments/total number of fragments) × 100.

*2.4. Statistical Analysis*

The data obtained for growth, yield, and nutrient content were subjected to statistical analysis using SAS software version 9.4 (SAS Institute, Cary, NC, USA) [43]. A two-way analysis of variance (ANOVA) was performed on the two factors of variety and treatment, to find the best combination between microorganisms and snap beans that enhanced the mineral nutrition and plant growth.

**3. Results**

*3.1. Plant Growth Parameters*

3.1.1. Shoot Dry Biomass

The results from this study indicate that the shoot dry biomass of the snap beans was significantly influenced by the different inoculations (Figure 1). Notably, the plants inoculated with *R. tropici* CIAT 899 in association with either *Serendipita indica* (C+*Si*) or *Rhizophagus irregularis* (C+*Ri*) had a significantly higher biomass compared with the other treatments. Conversely, the control group, which was not inoculated, produced the lowest biomass regardless of the stage of development. The analysis of variance confirmed that the double inoculation had a significant effect on the shoot dry biomass. Inoculating with the *Rhizobium tropici* strain CIAT 899 in association with *Serendipita indica* or *Rhizophagus irregularis* resulted in a significant improvement ($p \leq 0.001$) in the dry biomass of the aerial organs, both during the flowering stage and the pod stage (Table 2). Furthermore, a highly significant interaction ($p \leq 0.01$) between variety and treatment was observed at the flowering stage, where inoculation with *Serendipita indica* yielded the highest means in both the Contender and Garrafal Enana varieties. However, this interaction effect was not significant during the pod stage of development (Figure 1).

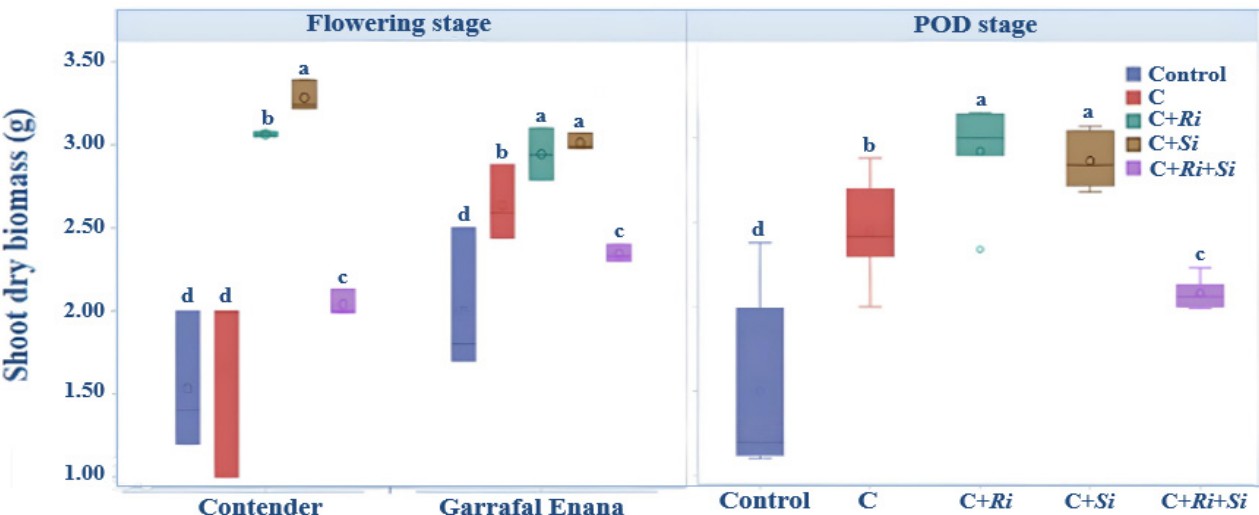

**Figure 1.** Box plots and Tukey HSD post-hoc test showing the effects of different inoculations on the above-ground biomass of snap bean at flowering and pod stages. Control: no inoculation; C: only *R. tropici* CIAT 899 inoculation; C+*Ri*: inoculation with *R. tropici* CIAT 899 and *Rhizophagus irregularis*; C+*Si*: inoculation with *R. tropici* CIAT 899 and *Serendipita indica*; C+*Ri*+*Si*: inoculation with *R. tropici* CIAT, *Rhizophagus irregularis*, and *Serendipita indica*. Within each graph, treatments that differ significantly are indicated with a different letter ($p \leq 0.05$). Midline shows the median, box shows the 25th and 75th percentiles, error bars show minimum and maximum values.

**Table 2.** ANOVA summary of mean response by treatment group using the Tukey test.

| Parameter Variables | | DF | F Value | (Pr > F) |
|---|---|---|---|---|
| SDB (at flowering) | Treat | 4 | 28.46 | <0.0001 |
| | Var | 1 | 32.13 | 0.02 |
| | Treat*Var | 4 | 4.91 | 0.008 |
| SDB (at harvest) | Treat | 4 | 32.48 | <0.0001 |
| | Var | 1 | 0.50 | ns |
| | Treat*Var | 4 | 2.84 | ns |
| RDB (at flowering) | Treat | 4 | 27.14 | <0.0001 |
| | Var | 1 | 6.20 | ns |
| | Treat*Var | 4 | 3.77 | 0.02 |
| RDB (at harvest) | Treat | 4 | 102.12 | <0.0001 |
| | Var | 1 | 1.01 | ns |
| | Treat*Var | 4 | 4.34 | 0.01 |
| NOD numbers (at flowering) | Treat | 4 | 55.04 | <0.0001 |
| | Var | 1 | 4.00 | ns |
| | Treat*Var | 4 | 3.16 | 0.04 |
| NOD numbers (at harvest) | Treat | 4 | 7.68 | 0.001 |
| | Var | 1 | 0.28 | ns |
| | Treat*Var | 4 | 3.30 | 0.03 |
| POD numbers | Treat | 4 | 177.06 | <0.0001 |
| | Var | 1 | 121.00 | 0.008 |
| | Treat*Var | 4 | 7.02 | 0.001 |

SDB: shoot dry biomass; RDB: root dry biomass; Treat: treatment; Var: variety; Treat*Var: interaction between treatment and variety; ns = not significant at the 5% level.

### 3.1.2. Root Dry Mass

The results for root dry mass are summarized in Table 3. Notably, *R. tropici* in association with *Serendipita indica* (C+*Si*) and *Rhizophagus irregularis* (C+*Ri*) demonstrated the highest biomass at both the flowering and pod stages. The increased biomass observed in plants subjected to double or triple inoculation, compared to non-inoculated plants, can be attributed primarily to the emergence of nodules on the roots and the proliferation of mycelial mass from symbiotic fungi. However, by the pod stage, the symbiotic relationship between *R. tropici* and the host plant had ceased, leading to a gradual disappearance and disintegration of the nodules within the soil. The analysis of variance revealed that the variety had no significant effect on the root dry biomass of the common bean (Table 2). Overall, the double inoculation significantly enhanced the root dry mass of snap bean, regardless of its developmental stage, with this effect being statistically significant ($p \leq 0.001$).

**Table 3.** The effect of rhizobial and mycorrhizal strains inoculation on root dry biomass and nodule numbers of common bean species.

| Treatments | | Root Dry Biomass at Flowering (g) | Root Dry Biomass at Harvest (g) | Nodule Number/Plant at Flowering | Nodule Number/Plant at Harvest | POD Number/Plant |
|---|---|---|---|---|---|---|
| | Control | 1.37 ± 0.47 d | 1.11 ± 0.15 d | 97 ± 5.21 e | 86 ± 0.47 b | 5 ± 0.002 d |
| | C | 2.39 ± 0.17 c | 2.80 ± 0.06 b | 251 ± 4.5 c | 92 ± 5.13 a | 9 ± 0.007 b |
| Contender | C+*Ri* | 2.76 ± 0.09 b | 2.97 ± 0.08 a | 273 ± 2.84 b | 91 ± 6.6 a | 11 ± 0.009 a |
| | C+*Si* | 3.05 ± 0.03 a | 2.98 ± 0.18 a | 295 ± 4.63 a | 87 ± 1.05 b | 12 ± 0.01 a |
| | C+*Ri*+*Si* | 2.37 ± 0.13c | 2.11 ± 0.17 c | 185 ± 5.29 d | 83 ± 6.8 c | 7 ± 0.006 c |
| | Control | 1.19 ± 0.06 d | 1.21 ± 0.02 d | 100 ± 1.52 d | 81 ± 2.12 c | 4 ± 0.002 d |
| | C | 2.47 ± 0.19 b | 2.43 ± 0.06 b | 253 ± 7.37 b | 93 ± 3 a | 11 ± 0.008 b |
| Garrafal Enana | C+*Ri* | 2.98 ± 0.12 a | 2.84 ± 0.01 a | 303 ± 3.18 a | 92 ± 7.09 a | 13 ± 0.01 a |
| | C+*Si* | 2.99 ± 0.01 a | 2.85 ± 0.02 a | 313 ± 2.96 a | 91 ± 1.41 ab | 13 ± 0.008 a |
| | C+*Ri*+*Si* | 2.33 ± 0.05 c | 2.15 ± 0.1 c | 199 ± 3.5 c | 90 ± 6.55 ab | 8 ± 0.003 c |

Control: no inoculation; C: only *R. tropici* CIAT 899 inoculation; C+*Ri*: inoculation with *R. tropici* CIAT 899 and *Rhizophagus irregularis*; C+*Si:* inoculation with *R. tropici* CIAT 899 and *Serendipita indica*; C+*Ri*+*Si*: inoculation with *R. tropici* CIAT 899, *Rhizophagus irregularis*, and *Serendipita indica*. Values are mean ± standard errors of replicates; (±): standard deviation from the average value presented (n = 6). Different letters within each column indicate significant differences ($p \leq 0.05$) using Tukey tests.

### 3.1.3. Number of Nodules

The MPN of autochthonous rhizobia ranged from 1397 to 9446 rhizobia per gram of soil, indicating a relatively low count. Interestingly, the introduction of *R. tropici*, whether alone or in combination with fungal strains, significantly increased the number of nodules compared with non-inoculated plants. Regardless of the bean variety, plants inoculated with *R. tropici* strain CIAT 899 exhibited the highest nodule counts at harvest time, as shown in Table 3. Additionally, the number of nodules tended to be higher during the flowering stage than the pod stage. Typically, nodule numbers peaked early in the flowering phase, remained relatively stable until approximately three weeks after flowering, and then declined significantly towards the end of the growth cycle.

The analysis of variance revealed that the variety had no significant effect on nodule count across different developmental stages, as outlined in Table 2. However, there was a noteworthy and statistically significant interaction ($p \leq 0.05$) between cultivar and treatment during both the flowering and harvest stages. The number of nodules appears to have been influenced more by the combined effects of the cultivar and the type of inoculation rather than being solely dependent on either the variety or the isolated effect of inoculation.

### 3.1.4. POD Number

The comparative analysis of different treatments reveals that the application of double inoculation, involving *R. tropici* CIAT 899 combined with either *Serendipita indica* or *Rhizophagus irregularis* fungi, led to a significant increase in pod yield. This is supported by statistical significance levels of $p \leq 0.001$, as shown in Tables 2 and 3. The Contender variety displayed a pod range of 5 to 12 pods per plant, while the Garrafal Enana variety exhibited a range of 4 to 13 pods. Notably, plants that remained uninoculated had the lowest pod counts.

Among the various inoculations, it is worth noting that plants inoculated with *R. tropici* CIAT 899 in association with *Rhizophagus irregularis* (C+Ri) achieved higher pod yield than non-inoculated plants. This increase in yield was independent of the variety, highlighting the significant impact of this specific double-inoculation treatment on pod production per plant.

The statistically significant enhancement in pod yield brought about by the combined effects of *Rhizobium tropici* and *Rhizophagus irregularis* underscores the potential of such symbiotic interactions in boosting agricultural productivity.

### 3.1.5. Percent Mycorrhization

The microscopic investigation of snap bean roots revealed notable cytological differences. In the absence of inoculation (as depicted in Figure 2a), the root systems of plants did not display any fungal structures. However, when mycorrhizal plants were stained with trypan blue, distinct structures characteristic of endomycorrhizal associations became evident.

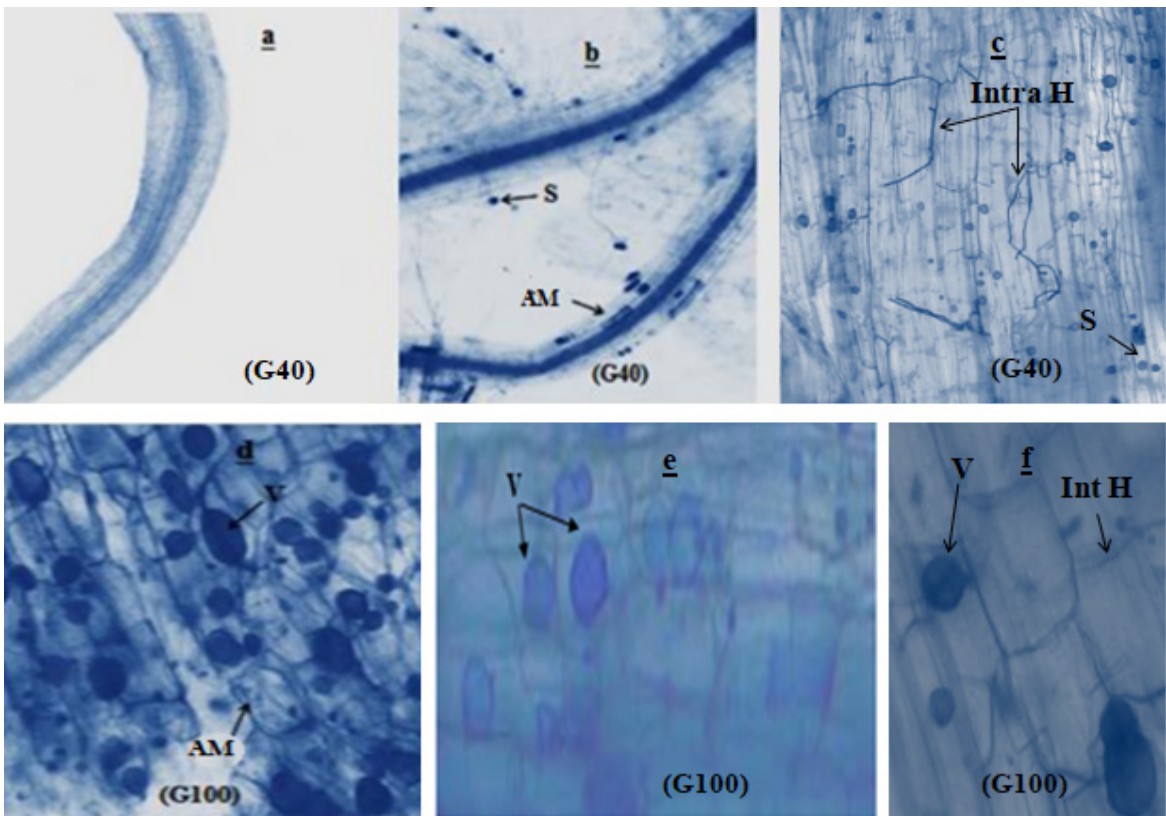

**Figure 2.** Microscopic observation of bean roots after 60 days of growth. (**a**): uninoculated snap bean root; (**b,e,f**): root inoculated with *Rhizophagus irregularis*; (**c**): root inoculated with *Serendipita indica*; (**d**): root co-inoculated with *Rhizophagus irregularis* and *Serendipita indica*. S: spore; V: vesicle; AM: arbuscular mycorrhizae; Int H: intercellular hyphae; Intra H: intracellular hyphae.

These mycorrhizal structures include spherical spores, intracellular hyphae (Figure 2b,c), arbuscular mycorrhizae, and oval vesicles interspersed between the cell cortexes (Figure 2d,e). Additionally, intercellular hyphae were observed branching along the root cortex (Figure 2f).

As the developmental stages progressed to flowering and fruiting, a notable expansion of infection spread was observed. This expansion in infection spread serves as confirmation of the successful establishment of arbuscular mycorrhizal fungi and the endophyte within the root systems. Remarkably, the colonization patterns indicated robust infection by *Serendipita indica*, *Rhizophagus irregularis*, and their combined association, as depicted in Figure 3. Specifically, the colonization by these fungi was strikingly extensive, underscoring the effectiveness of their interaction. This is particularly evident when considering the combined association of the two strains. At the flowering stage, the colonization rate by this association did not surpass 65%, reflecting the ongoing establishment process. However, by the time of harvest, the colonization rate exceeded 80%, indicating the progressive success of this combined inoculation strategy in promoting fungal colonization within the root systems. Only inoculation with fungal strains showed a highly significant effect ($p \leq 0.001$) at both the flowering and podding stages.

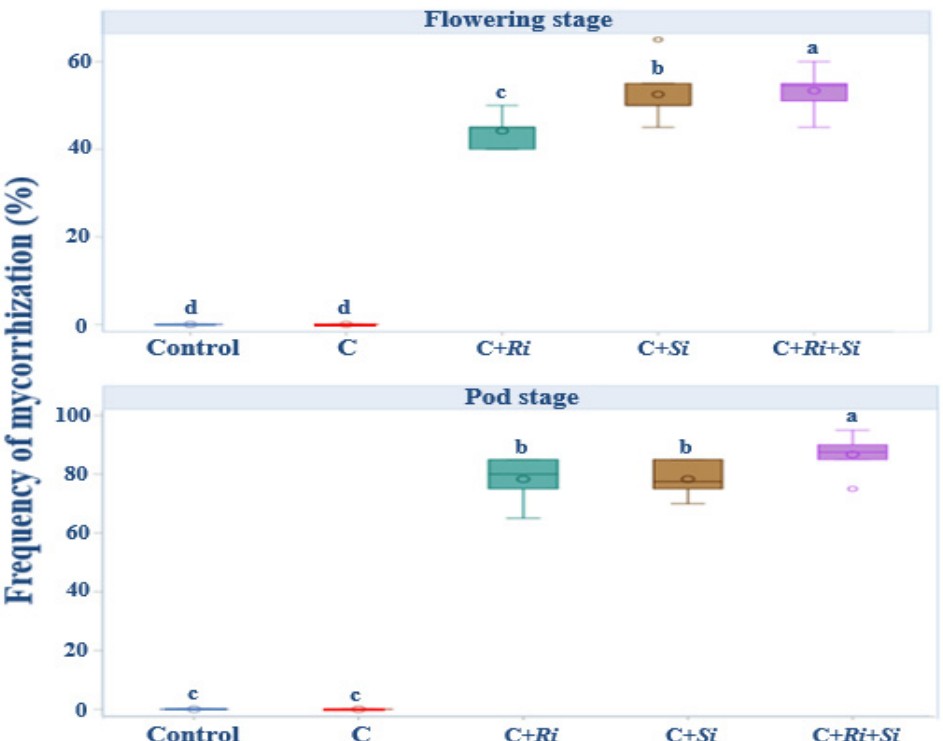

**Figure 3.** Box plots and Tukey HSD post-hoc test showing the frequency of mycorrhizal colonization. Control: no inoculation; C: only *R. tropici* CIAT 899 inoculation; C+*Ri*: inoculation with *R. tropici* CIAT 899 and *Rhizophagus irregularis*; C+*Si*: inoculation with *R. tropici* CIAT 899 and *Serendipita indica*; C+*Ri*+*Si*: inoculation with *R. tropici* CIAT 899, *Rhizophagus irregularis*, and *Serendipita indica*. Within each graph, treatments that differ significantly are indicated with a different letter ($p \leq 0.05$). Midline shows the median, box shows the 25th and 75th percentiles, error bars show minimum and maximum values.

### 3.1.6. Nitrogen Content

The mean nitrogen concentrations in leaves and roots during the flowering and harvesting stages are depicted in Figure 4. It is noteworthy that uninoculated plants consistently exhibited nitrogen concentrations below or close to 0.15%, regardless of the stage of growth, both in leaves and roots. In contrast, the introduction of *R. tropici* CIAT 899 inoculation or symbiotic fungi resulted in a noticeable increase in nitrogen levels in both leaves and roots. This increase in nitrogen availability indicates the positive

impact of these inoculation strategies on nutrient acquisition by the plants. The results highlight the importance of symbiotic associations and inoculation practices in enhancing nitrogen uptake and utilization. While uninoculated plants struggled to maintain nitrogen concentrations above 0.15%, the introduction of beneficial microorganisms significantly improved nitrogen availability. The analysis of variance in Table 4 demonstrates that there were significant differences among the treatments ($p \leq 0.001$), although the differences between bean varieties were not found to be significant at the 5% level.

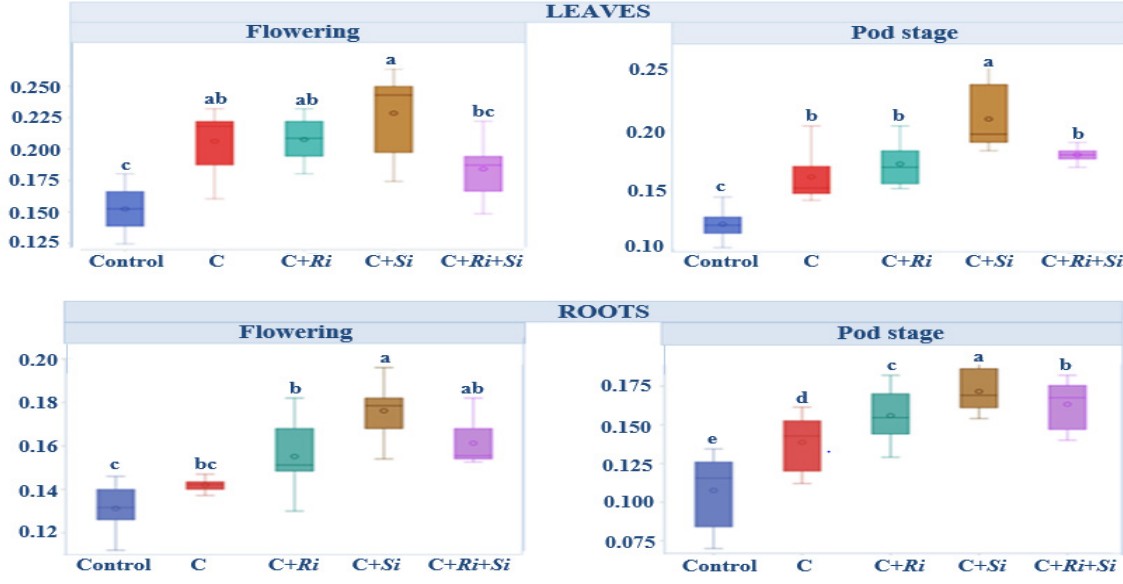

**Figure 4.** Box plots and Tukey HSD post-hoc test showing the nitrogen content per plant in bean leaves and roots at the flowering and pod stages. Control: no inoculation; C: only *R. tropici* CIAT 899 inoculation; C+*Ri*: inoculation with *R. tropici* CIAT 899 and *Rhizophagus irregularis*; C+*Si:* inoculation with *R. tropici* CIAT 899 and *Serendipita indica*; C+*Ri*+*Si*: inoculation with *R. tropici* CIAT 899, *Rhizophagus irregularis*, and *Serendipita indica.* Within each graph, treatments that differ significantly are indicated with a different letter ($p \leq 0.05$). Midline shows the median, box shows the 25th and 75th percentiles, error bars show minimum and maximum values.

3.1.7. Phosphorus Content

The phosphorus content is illustrated in Figure 5. Notably, during the flowering stage, plants inoculated with *Serendipita indica* and *R. tropici* exhibited higher phosphorus content in leaves (0.82 mg/g dry matter, DM) and roots (0.047 mg/g DM) compared with non-inoculated plants. Subsequently, at the harvest stage, these inoculated plants had the highest phosphorus content, averaging 1.89 mg/g DM in leaves and 0.108 mg/g DM in roots. The analysis of variance, presented in Table 4, focused on the phosphorus content in aerial organs and roots during the flowering and pod stages. Specifically, inoculation with the *Rhizobium tropici* strain CIAT 899, either alone or in combination with symbiotic fungi, resulted in increased phosphorus levels during the flowering stage. This increase reached significance ($p \leq 0.05$) in leaves and was highly significant ($p \leq 0.01$) in roots. Notably, the interaction between variety and treatment did not have a significant effect. Overall, throughout the developmental stages, it is evident that single, double, or triple-inoculation strategies led to higher phosphorus content in leaves compared with roots. Additionally, significant differences between treatments ($p \leq 0.001$) were observed in both leaves and roots during the flowering and pod stages.

**Table 4.** ANOVA summary of mean response by treatment group for phosphorus and nitrogen uptake, according to Tukey test.

| Parameters | Variables | DF | F Value | (Pr > F) |
|---|---|---|---|---|
| Leaves phosphorus content (flowering stage) | Treat | 4 | 21.70 | <0.0001 |
| | Var | 1 | 0.01 | ns |
| | Treat*Var | 4 | 0.47 | ns |
| Leaves phosphorus content (at harvest) | Treat | 4 | 40.55 | <0.0001 |
| | Var | 1 | 0.22 | ns |
| | Treat*Var | 4 | 2.95 | ns |
| Root phosphorus content (flowering stage) | Treat | 4 | 30.35 | <0.0001 |
| | Var | 1 | 4.95 | ns |
| | Treat*Var | 4 | 2.92 | ns |
| Root phosphorus content (at harvest) | Treat | 4 | 39.13 | <0.0001 |
| | Var | 1 | 2.67 | ns |
| | Treat*Var | 4 | 3.77 | ns |
| Leaves nitrogen content (flowering stage) | Treat | 4 | 9.08 | 0.0005 |
| | Var | 1 | 0.04 | ns |
| | Treat*Var | 4 | 0.34 | ns |
| Leaves nitrogen content (at harvest) | Treat | 4 | 22.86 | <0.0001 |
| | Var | 1 | 17.64 | 0.05 |
| | Treat*Var | 4 | 1.28 | ns |
| Root nitrogen content (flowering stage) | Treat | 4 | 18.93 | <0.0001 |
| | Var | 1 | 37.64 | 0.02 |
| | Treat*Var | 4 | 5.11 | 0.007 |
| Root nitrogen content (at harvest) | Treat | 4 | 10.88 | 0.0002 |
| | Var | 1 | 7.29 | ns |
| | Treat*Var | 4 | 0.63 | ns |

Treat: treatment; Var: variety; Treat*Var: interaction between treatment and variety; ns = not significant at the 5% level.

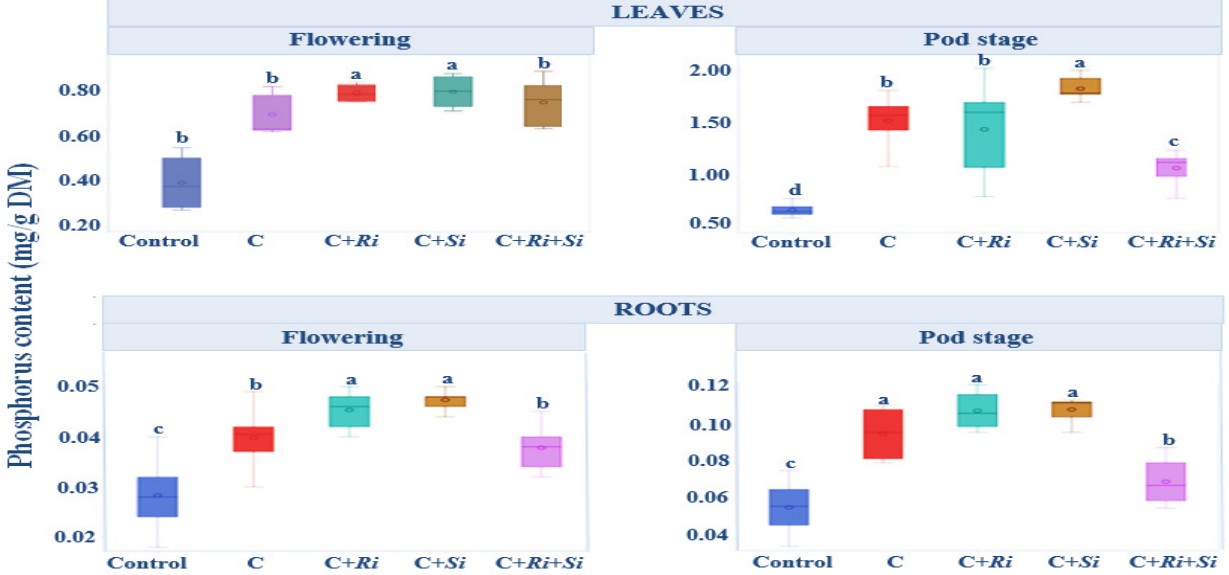

**Figure 5.** Box plots and Tukey HSD post-hoc test showing the leaf and root phosphorus levels of *Phaseolus vulgaris* L. at the flowering and pod stages. Control: no inoculation; C: only *R. tropici* CIAT 899 inoculation; C+*Ri*: inoculation with *R. tropici* CIAT 899 and *Rhizophagus irregularis*; C+*Si*: inoculation with *R. tropici* CIAT 899 and *Serendipita indica*; C+*Ri*+*Si*: inoculation with *R. tropici* CIAT 899, *Rhizophagus irregularis*, and *Serendipita indica*. Within each graph, treatments that differ significantly are indicated with a different letter ($p \leq 0.05$). Midline shows the median, box shows the 25th and 75th percentiles, error bars show minimum and maximum values.

## 4. Discussion

As important tools for sustainable agriculture, mycorrhizae and rhizobacteria are used to improve plant productivity and reduce chemical inputs. The present study was conducted to explore the potential of beneficial microorganisms for improving the productivity of common bean crops.

In Tunisia, the yield of common beans is constrained by the lack of phosphorus availability and the decline in native rhizobia that fix atmospheric nitrogen. The study conducted provides valuable insights into how different snap bean varieties respond to inoculation with rhizobia and symbiotic fungi. The findings demonstrate that inoculation effectively promotes plant growth, with the most pronounced effects observed during the fruiting stage when plants have a heightened demand for essential mineral elements. In fact, the combined inoculation of *R. tropici* CIAT 899 with *Serendipita indica* and *Rhizophagus irregularis* fungi enhanced both aerial (Figure 1) and root dry mass (Table 3), although the results remain comparable to single inoculations. However, accurately predicting plant responses to mycorrhizal inoculation poses a challenge due to the complex nature of interactions between plants and mycorrhizal fungi [44]. This intricate interplay is dependent on various factors, including plant species, fungal strains, the developmental stage of the host plant, and environmental conditions [45].

Notably, there is a strong affinity between *Rhizobium tropici* CIAT 899 and *Serendipita indica*, as evidenced by the highest biomass and pod yields achieved in the two varieties. These findings align with recent research investigating plants that were co-inoculated with plant-growth-promoting rhizobacteria (PGPRs) and root endophytic fungi [46–48]. Numerous studies on bio-stimulants have confirmed their positive impact on plant growth. These bio-stimulants increase production and yield while improving the nutritional quality of the plants [49]. Specifically, they have been found to increase the protein, flavonoid, and polyphenols content in snap bean [3] and faba bean seeds [50–52].

A noteworthy outcome of this study is that plants inoculated with *Serendipita indica* and/or *Rhizophagus irregularis* exclusively exhibited fungal structures (Figure 2). The absence of these structures could be due to the lack of fungi or a decrease in microbial populations caused by excessive pesticide use [53]. In semi-arid zones, the soils often have a limited number of ineffective rhizobia populations for Fabaceae plants [54]. Edaphic factors play a significant role in influencing the mycorrhizal fungi spores' density and root colonization [55]. Previous studies suggest that pH and organic matter content regulate the sporulation of arbuscular mycorrhizal fungi [56]. Many researchers have demonstrated that the *Rhizophagus irregularis* genus is typically found in soils with neutral or alkaline pH [57–59]. This aligns with the promising results observed in this study, where the inoculation of bean varieties with *Rhizophagus irregularis* resulted in positive outcomes. Indeed, inoculated plants revealed improvements across various parameters, particularly growth and mineral nutrition. The effectiveness of the *Rhizobium tropici* strain CIAT 899 and the symbiotic fungi used is demonstrated. There is clear potential for authentic symbiotic fungi strains to act as biofertilizers and enhance mineral nutrition. The inoculation of snap beans with *R. tropici* in combination with *Serendipita indica* or *Rhizophagus irregularis* leads to favorable results for phosphorus uptake (Figure 5). This mutually beneficial association gradually develops and thrives as accessible nutrients directly support the growth of plant roots in the soil. *Serendipita indica* and *Rhizophagus irregularis* are known for their effective assimilation of phosphorus. Meiyan et al. [59] demonstrated that mycorrhizal fungi, once they reach a minimum threshold of rhizosphere invasion, play a crucial role in phosphate nutrient assimilation as transport mediators. Robust evidence has shown that *Serendipita indica* improves acid and alkaline phosphatase activities, leading to increased phosphorus uptake in rice and rapeseed [60,61]. *Serendipita indica* not only boosts soil phosphatase activities but also improves phosphorus absorption in *Poncirus trifoliata* by increasing the expression of phosphate transporter genes (PT3, PT5, and PT6) [62]. The endophyte *Serendipita indica* has the potential to enhance phosphorus uptake and plant growth by mobilizing inorganic phosphorus and stimulating phosphatase enzyme activ-

ity [63,64]. Indeed, enhanced phosphorus levels have a significant impact on not just plant growth but also nodulation and biological nitrogen fixation [65,66]. Thus, the inoculation with *R. tropici* CIAT 899, either alone or with *Serendipita indica* or *Rhizophagus irregularis*, increased nitrogen content (Figure 4) compared with the control. Furthermore, the increased phosphorus content resulted in higher numbers of nodules and greater root dry mass in comparison to the control group (Table 3). This suggests an improvement in symbiotic N$_2$ fixation and/or nitrogen uptake, potentially reducing the need for excessive nitrogen fertilizer application [67–69]. Additionally, Sheramati et al. [70] discovered that Serendipita indica enhanced the growth of Arabidopsis and tobacco seedlings by promoting nitrogen accumulation and increasing the expression of genes related to nitrogen reductase and starch-degrading enzyme glucan–water dikinase in their roots. This enzyme plays a crucial role in nitrogen and starch metabolism in seedlings, which are essential for their growth and development. Numerous studies have highlighted that in situations of nitrogen deficiency, the secretion of root exudates triggers a molecular communication that culminates in symbiosis between soil bacteria and host plants [71,72]. This intricate process entails the activation of Nod genes, which are pivotal in the synthesis of Nod factors. The growth of nodules plays a crucial role in the activation of genes that influence nitrogen and carbon metabolism, protein expression, and the production of secondary metabolites. Ultimately, these mechanisms fortify the plant's defense system [73].

In the future, to obtain a more comprehensive understanding, we can undertake functional studies on genes associated with nitrogen and phosphorus transport, transcription factors, phytohormones, and the signaling pathway of mycorrhizal symbiosis. Furthermore, we can also examine physiological parameters such as phenol content and enzymes linked to these processes.

## 5. Conclusions

In conclusion, this study sheds light on the dynamic interactions between bean varieties, rhizobia, and symbiotic fungi. Inoculation positively impacts plant growth, particularly during the fruiting stage, addressing mineral requirements. Mycorrhizal symbiosis enhances nutrient uptake, potentially reducing symbiotic reliance under nutrient-rich conditions. Unique fungal structures were observed in *Serendipita indica* and/or *Rhizophagus irregularis*-inoculated plants, potentially influenced by microbial populations and edaphic factors. The impressive phosphorus assimilation abilities of authentic symbiotic fungi strains such as *Serendipita indica* and *Rhizophagus irregularis* make them promising biofertilizers. The combination of *R. tropici* CIAT 899 with these symbiotic fungi enhances nodule numbers, thereby improving nitrogen fixation potential. Inoculating fungi into the soil not only promotes plant growth but also balances the soil by decomposing organic matter and transforming it into humus. However, predicting the effects of mycorrhizal symbiosis remains complex, and the physiological mechanisms underlying growth induction need further exploration. Overall, these findings can contribute to sustainable agricultural strategies and enhance crop performance. However, comprehensive information on nutrient transport in *Serendipita indica* remains scarce.

**Author Contributions:** Conceptualization, H.B., M.M. (Mongi Melki) and M.M. (Mouna Mechri); methodology, W.S. and A.O.; software, H.B. and R.H.; validation, C.C. and T.R.; formal analysis, H.B, M.M. (Mouna Mechri) and M.M. (Mongi Melki); resources, M.M. (Mouna Mechri); data curation, M.M.; writing—original draft preparation, H.B.; writing—review and editing, H.B., K.B., T.R., A.H. and E.F.A.; visualization, M.M. (Mouna Mechri); supervision, C.C.; project administration, E.F.A.; funding acquisition, E.F.A. All authors have read and agreed to the published version of the manuscript.

**Funding:** The authors would like to extend their sincere appreciation to the Researchers Supporting Project Number (RSP2023R134), King Saud University, Riyadh, Saudi Arabia.

**Data Availability Statement:** Data are contained within the article.

**Acknowledgments:** The authors would like to extend their sincere appreciation to the Researchers Supporting Project Number (RSP2023R134), King Saud University, Riyadh, Saudi Arabia.

**Conflicts of Interest:** The authors declare no conflict of interest.

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
