# Peer review of "Synergistic Interaction of Rhizobium tropici, Rhizophagus irregularis and Serendipita indica in Promoting Snap Bean Growth"

_agronomy, doi:10.3390/agronomy13102619_

Round 1

Reviewer 1 Report

I appreciate the well written manuscript entitled " Combined inoculation of Serendipta indica, Rhisophagus irregularis and CIAT 899 tropici as biostimulants to improve growth and nutrient bioavaibility of common bean”. The manuscript describes a very interesting study. The authors did a lot of work. The subject is both interesting and worth publishing in “Agronomy”. That being said, the manuscript has the potential to be accepted. However, there is still some issues need to be addressed before the paper could be accepted attached to my report. Kind Regards.

Minor editing of English language required

Author Response

# Reviewer 1

Dear professor,

We wanted to express our sincere appreciation for the time and effort you dedicated to reviewing and correcting our article. We sincerely apologize for the errors made in the manuscript. Considering your suggestion, we have incorporated the revised sentence into a new version of the manuscript. Your corrections and enhancements have greatly improved the quality of the content.

We genuinely hope that these revisions and enhancements will meet your expectations.

Comment 1: The title is very long. I highly recommended that the author rework on the title to be more attractive in terms of what the readers will see new in this manuscript.

Response 1: The correction has been made.

Synergistic Interaction of Rhizobium, Serendipidata indica and Rhizophagus irreguaris in Promoting Snap Bean Growth.

Comment 2: The presentation of the key findings of experimental results should be improved and data regarding the mainly measured indicators should be presented. Please focus on the novelty of this work.

Response 2: The correction has been made (Please see p.1 abstract).

Comment 3: keywords are already stated in the title. Please consider changing the keywords list and use synonyms.

Response 3: Key words: Mycorrhizae; endophytic fungus; mineral nutrition; plant-microbes interactions.

The correction has been made (Please see p.1 line 45).

Comment 4: Revise “Organic matter ismost import carbon fraction” …

Response 4:  Sentence has been changed.

 As it's known that organic matter, specifically the carbon fraction, plays a critical role in preserving soil health.

The correction has been made (Please see p.2 line 73).

Comment 5: Subscript “N2”

Response 5:  « N». The correction has been made (Please see p.3 line 95).

Comment 6: Revise 'yiela'.

Response 6:  This sentence has been removed.

Comment 7: The authors are recommended to add more details on the significance of the study and provide a hypothesis of the present study at the end of the introduction to give the reader more information regarding the purpose and the mechanistic used to achieve this goal, then may refer some lack in the previous study regarding some aspects.

Response 7:  Thank you for your thoughtful review and your request for more information regarding the purpose of our work.

The correction has been made (Please see p.3 line 112-121).

Comment 8: Give some details about the pots such as (Height, Diameter, etc..)

Response 8:  The pots used for the experiment had a height of 20 cm and a diameter of 25 cm at the top and 19 cm at the base.

 The correction has been made (Please see p.3 line 137).

Comment 9: By which method???

Response 9:  The soil analysis was conducted at the regional laboratory. pH was measured in a soil/ water suspension (1/2.5) by potentiometry using a calibrated pH-meter.

Comment 10: Concentration???

Response 10:  The concentration of sodium hypchlorite solution is 3%.

Surface sterilization was performed by immersion of seeds for 1 min in ethanol (70%), which were then immersed for 10 min in 3 % of NaOCl, respectively.

The correction has been made (Please see p.4 line 146).

Comment 11: How did you measure the density of the cell in the culture media???

Response 11:  The population density of bacteria in Yeast Mannitol Broth was estimated using the standard plate count' method. A dilution series was prepared, and the number of colonies counted on a plate provided the colony forming units (CFU).

Comment 12: Give the full name of YEM.

Response 12:  Yeast Extract Mannitol.

The correction has been made (Please see p.4 line 164).

Comment 13:

13.1. How ??? see https://doi.org/10.1016/j.micres.2022.127254

13.2. How much mycorrhizal spores in 50 g inoculum?? How did the authors prepare the mycorrhizal inoculum???

Response 13: Thank you for your thoughtful review and your request for more detailed information on the research methodology. According to our understanding we try to answer. I hope we revised correctly.

13.1.  In the mentioned article, the author utilized the mycorrhizal fungus Funneliformis constrictum. This strain was isolated from the soil and prepared by the researchers themselves prior to its application. The quantity used, 5g per pot, corresponds to approximately 50 spores per gram. It is worth noting that Duc et al. 2023, have used 50g of inoculum for Funneliformis, which translates to around 30 spores per gram. The variation in quantities employed may be influenced by many factors such as the quality of the inoculum, the specific strain utilized, plant species, and the prevailing environmental conditions.

Duc et al. 2023 : https://doi.org/10.1016/j.plaphy.2023.107892.

3.2. Rhizophagus irregularis previously called Glomus intraradice used in this study was a commercial inoculum ready to use. We sold it, we didn't prepare the inoculum ourselves. For couting AMF spores from Glomus inoculum, we used the wet-sieving technique. 

Fifty grams of inoculum was thoroughly mixed into each pot by the layering method (Jackson et al.,1972); that received approximately 1000 spores (20 spores gram-1).

Tajini et al., 2012 who worked with the same strain Glomus intraradice, used also 50 grams of AMF inoculum which received approximately 1000 spores of the AMF species contained at least 20 infective propagules of AMF per gram of chopped root.

  • doi: 10.1016/j.sjbs.2011.11.003. Epub 2011

Jafari et al., 2018 used 50 grams of Glomus intraradice to inoculate alfalfa seedlings.

  • doi: 10.1016/j.aninu.2017.08.008.

Comment 14: Provide the full name of your treatments in the figure caption to be easy for the readers.

Response 14:  The correction has been made (Please see p 6.line 236).

Comment 15: Where are the alphabetical letters of the statistics???

Response 15:  The correction has been made (Please see p.8 Table 3).

Comment 16: Provide the full name of your treatments in the table caption to be easy for the readers.

Response 16:  The correction has been made (Please see p.8 line 286).

Comment 17: What about the mycorrhizae already present in the soil?? Did the authors sterilize the soil???

Response 17:  We observed the absence of mycorrhizae in our soil samples, which led us to not discuss the competition between indigenous and allochthonous strains. This absence could potentially be attributed to soil disturbances, such as intensive agricultural practices, excessive tilling, or the application of specific chemical fertilizers or pesticides. These factors have the potential to disrupt or hinder the growth of mycorrhizal fungi.

Comment 18:  The authors should cite tables and figures in the discussion section to keep the reader constantly linked to the results.

Response 18:  The correction has been made (Please see the discussion).

Comment 19: Subscript (N2)

Response 19:  The correction has been made (Please see p.14 line 424).

Comment 20: Subscript (N2).

Response 20:  The sentence has been delated.

We sincerely appreciate your insightful feedback.

Reviewer 2 Report

The manuscript entitled “Combined inoculation of Serendipta indica, Rhisophagus irregularis and CIAT 899 tropici as biostimulants to improve growth and nutrient bioavaibility of common bean” fits with the general scope of the Journal Agronomy MDPI. In this work, the authors investigated the potential use of different microbial biostimulant for increasing growth and nutrient availability on common bean plants. The work could be very interesting for researchers and farmers. The manuscript is well written and organized. I have some minor comments that need to be addressed before the publication.

Abstract

The abstract is well structured and written. I suggest to describe the results with percentage (increase or decrease compared to the control).

Introduction

Introduction is well written, I have few comments on it.

Line 50: never start a phrase with “and”.

Lines 56-59: rephrase, too long

Line 64: “Organic matter ismost import carbon fraction”, I think that there is a typo in this phrase, please correct it.

Lines 71-77: these phrases need references.

Line 80: “avaialbilit” is a typo.

Line 96: “yiela” is a typo

Materials and methods

Materials and methods section is well written and organized. All information are clearly presented. I have only few comments on it.

Line 126: please start the phrase with “Ten”.

Line 187: “thirty”.

Line 192: data … WERE subjected to…

Results

Results are clearly presented. Only few suggestions:

Please improve the quality of the Figures 1, 3, 4 and 5.

Line 213: I believe that “overage” is a typo.

I think that for ameliorating the readability of the table 3 and of the figure, authors need to include the letters for the means separation.

Discussions

The discussions section is well written, I suggest to the authors to include more reference since there are some parts that are lacking.

Line 361: “Serendipita” in italics.

Conclusions are good.

References

Please carefully check the references, there are some formatting issues.

Other general comments:

Paragraph titles must be in italics.

Author Response

# Responses to Reviewer 2:

Dear professor,

Thank you for your invaluable feedback. We sincerely apologize for the errors made in the manuscript. Considering your suggestion, we have incorporated the revised sentence into a new version of the manuscript. We genuinely hope that these revisions and enhancements will meet your expectations.

Comments and Suggestions for Authors

The manuscript entitled “Combined inoculation of Serendipta indicaRhisophagus irregularis and CIAT 899 tropici as biostimulants to improve growth and nutrient bioavaibility of common bean” fits with the general scope of the Journal Agronomy MDPI. In this work, the authors investigated the potential use of different microbial biostimulant for increasing growth and nutrient availability on common bean plants. The work could be very interesting for researchers and farmers. The manuscript is well written and organized. I have some minor comments that need to be addressed before the publication.

Comment 1: Abstract

The abstract is well structured and written. I suggest to describe the results with percentage (increase or decrease compared to the control).

 Response 1: Thank you for your feedback and taking the time to review our abstract thoroughly, we have revised the abstract to make the message clear. Changes has been made in the revised manuscript.

Comment 2: Introduction

Introduction is well written, I have few comments on it.

# Comment 2.1: Line 50: never start a phrase with “and”.                                                                                  Response 2.1: The correction has been made (please see P.2 line 64).

# Comment 2.2: Lines 56-59: rephrase, too long: “The advancement in the application of fungi and rhizobacteria in agriculture represents a transformative development in modern farming pratices.”

Response 2.2: The use of these microorganisms is a significant advancement in modern farming practices.

The correction has been made (please see p.2 line 70).

# Comment 2.3: Line 64: “Organic matter ismost import carbon fraction”, I think that there is a typo in this phrase, please correct it.

Response 2.3: The correction has been made (please see p.2 line 73-74 ).

# Comment 2.4: Lines 71-77: these phrases need references.

Response 2.4: The correction has been made (please see p.2 line 83-85).

# Comment 2.3: Line 80: “avaialbilit” is a typo.

Response 2.3It should be ‘’ availability ‘’                                                                                                     The correction has been made (please see p.3 line 91).

# Comment 2.4: Line 96: “yiela” is a typo

Response 2.4: It should be “yield”

The correction has been made (please see p.3 line 111 ).

Comment 3: Materials and methods

Materials and methods section is well written and organized. All information are clearly presented. I have only few comments on it.

# Comment 3.1: Line 126: please start the phrase with “Ten”.

Response 3.1: The correction has been made (please see p.4 line 148).

# Comment 3.2: Line 187: “thirty”.

Response 3.2: The correction has been made (please see p.5 line 211).

# Comment 3.3: Line 192: data … WERE subjected to…

Response 3.3: The correction has been made (please see p.5 line 217).

Comment 4: Results

Results are clearly presented. Only few suggestions:

#Comment 4.1: Please improve the quality of the Figures 1, 3, 4 and 5.

Response 4.1: The correction has been made (please see the Figures 1, 3, 4 and 5).

# Comment 4.2: Line 213: I believe that “overage” is a typo.

Response 4.2: The correction has been made (please see p.6 line 237).

# Comment 4.3: I think that for ameliorating the readability of the table 3 and of the figure, authors need to include the letters for the means separation.

Response 4.3: Dear reviewer, we have taken note of your observation regarding the clarity of table 3 and of the figure. We apologize for any difficulty you encountered in reading them. In response, we will ensure that the letters for the means separation are presented in the revised table and all Figures.

The correction has been made (please see Table 3 and Figures 1, 3, 4 and 5).

# Comment 5: Discussions

# Comment 5.1: The discussions section is well written, I suggest to the authors to include more reference since there are some parts that are lacking.

Response 5.1: The correction has been made (please see the discussion part).

# Comment 5.2: Line 361: “Serendipita” in italics.

Response 5.2: This sentence has been delated.

Comment 6: Conclusions are good.

Comment 7: References

Please carefully check the references, there are some formatting issues.

Response 7: Upon revisiting our references, we have carefully checked the style to ensure its consistency and adherence to the required format. We have arranged the citations and references according to the journal requirement in the revised manuscript.

Comment 8: Other general comments:

Paragraph titles must be in italics.

The correction has been made (please see the titles).

Reviewer 3 Report

1) The text requires extensive writing and English usage revision.    For example:    the phrase ", globally common bean..." (lines 44 and 45) is incorrectly started with a comma;    "Quility" (line 53), "ismost import" (line 64), "have stdudies" (line 90), "yela" (line 96), "flowring" (table 3); "results shows" (line 269), and several other words are spelled incorrectly.    In many interrupted sentences in the introduction, commas should be included. For example: "After the green revolution world" (lines 54); "...and symbiotic fungi" (line 84), and many others.    I do not understand the meaning of the sentence "...agricultural yield resultantly chemical..." (lines 55 and 56).    "In response symbiotic" (line 77) should be written as "in response to...".   Please revise the entire manuscript accordingly.     2) In the Abstract, I recommend rewriting and improving the following sentence to better clarify what the authors intended to communicate: "This research work suggested that the use of these symbiotic microorganisms into degraded soils could be an overall functionality of the agro-ecosystem."     3) In the Introduction and Discussion, I recommend incorporating better paragraph separation. They are too long and interrupted. Also, when discussing quantitative information on Tunisia, I recommend introducing this topic better, considering that the authors did not provide quantitative information on other countries.     3.1) There are several statements without citations (see lines 67-69; lines 71-78). Please review and revise the entire manuscript accordingly.    4) I recommend rewriting the topic title "2.3 Parameters to be studied." In this title version, it seems the parameters will be studied in the future. It would be better to write "Parameters" or "Studied Parameters."    5) Regarding the results, the resolution quality of the Figure 1 should be improved.     5.1) I recommend providing more details on statistics significance, differences and tests within the result figures and result (by using legends or captions).    5.2) Many comments provided in the results should be removed or placed in the discussion. For example: lines 259-262; lines 295-297.    6) Concerning beans, other articles on this research topic have provided insights into biochemical parameters in addition to growth parameters. Within this context, I recommend that the authors provide perspectives on biochemical and molecular parameters for future studies in the discussion. Please refer to the following articles (these are just a few examples; there are many others):    https://www.mdpi.com/2073-4395/10/2/189   https://www.mdpi.com/2076-3417/12/2/776   “ How Does the Addition of Biostimulants Affect the Growth, Yield, and Quality Parameters of the Snap Bean (Phaseolus vulgaris L.)? How Is This Reflected in Its Nutritional Value?”    By the way, I recommend that the authors read and cite these articles to better clarify the novelty of the current research in the introduction and discussion and to provide readers with a better overview of current research in this topic.     6.1) The authors mentioned molecular advances (line 67). Have they provided any molecular insight? If yes, it should be incorporated into the current manuscript.

Please see my comments to the authors.

Author Response

# Responses to Reviewer 3:

Dear professor,

We wanted to express our sincere appreciation for the time and effort you dedicated to reviewing and correcting our article. We sincerely apologize for the errors made in the manuscript. Considering your suggestion, we have incorporated the revised sentence into a new version of the manuscript. Your corrections and enhancements have greatly improved the quality of the content.

We genuinely hope that these revisions and enhancements will meet your expectations.

  • The text requires extensive writing and English usage revision.   

Dear reviewer, we have taken note of your observation regarding the English usage revision. We apologize for any difficulty you encountered in reading them. In response, we will ensure to revise the entire article and address the requested corrections.

For example:   

  1. the phrase ", globally common bean..." (lines 44 and 45) is incorrectly started with a comma;   

Response: the sentence has been changed (please see p.2 line 56).

  1. "Quility" (line 53).

Response: the sentence has been changed (please see p.2 line 60).

  • "ismost import" (line 64).

Response: the sentence has been changed (please see p.2 line 73-74 ).

  1. "have stdudies" (line 90).

Response: the sentence has been changed (please see p.3 line 101-103).

  1. "yela" (line 96).

Response: the sentence has been delated.

  1. "flowring" (table 3).

Response: "flowering" the correction has been made (please see p.8 Table 3).

  • "results shows" (line 269).

Response: the sentence has been changed (please see p.9 line 299).

  • … and several other words are spelled incorrectly.   

Response: Thank you for your feedback and taking the time to review our article thoroughly, we have revised it to make the message clear. Changes has been made in the revised manuscript.

  1. In many interrupted sentences in the introduction, commas should be included.

For example: "After the green revolution world" (lines 54);

            Response: the sentence has been changed (please see p.2 line 58-60).

             "...and symbiotic fungi" (line 84), and many others.   

               Response: the sentence has been delated.

 I do not understand the meaning of the sentence "...agricultural yield resultantly chemical..." (lines 55 and 56).   

               Response: the sentence has been delated.

 "In response symbiotic" (line 77) should be written as "in response to...".  

                  Response: the sentence has been changed (please see p.2 line 88-90).

  1. Please revise the entire manuscript accordingly. 

We have diligently made corrections to the article and have taken into account your valuable feedback. We genuinely hope that you will find the revised version to be significantly improved and that it addresses all the concerns and suggestions you previously mentioned. Thank you for your time and consideration.

  • In the Abstract, I recommend rewriting and improving the following sentence to better clarify what the authors intended to communicate: "This research work suggested that the use of these symbiotic microorganisms into degraded soils could be an overall functionality of the agro-ecosystem." 

Response: This sentence has been delated and the abstract has been changed. (Please see abstract p.1).

  • In the Introduction and Discussion, I recommend incorporating better paragraph separation. They are too long and interrupted.

Response: We certainly appreciate your insight. We endeavored to edit the introduction and discussion parts.

Comment: Also, when discussing quantitative information on Tunisia, I recommend introducing this topic better, considering that the authors did not provide quantitative information on other countries.   

Response: we highlighted the instance of pesticide usage in Tunisia and globally, bringing attention to its persistent expansion over time. (Please see p.2 lines 57-66).

3.1. Comment: There are several statements without citations (see lines 67-69; lines 71-78). Please review and revise the entire manuscript accordingly.    

Response:  Thank you for the comments regarding citation, we have added thirty-five citations in the revised manuscript.

  • Sixteen citations have been added to the introduction.
  • Five citations have been added to the material and method section.
  • Seventeen citations have been added to the discussion.

Reference for lines 67-69 has been added (Please see p. 2 line 82)

      Reference for lines 71-78 has been added (Please see p. 3 line 112)

.

  • I recommend rewriting the topic title "2.3 Parameters to be studied." In this title version, it seems the parameters will be studied in the future. It would be better to write "Parameters" or "Studied Parameters."    

Response: The correction has been made (please see p.13 line 188).

  • Regarding the results, the resolution quality of the Figure 1 should be improved. 

Response: The correction has been made (please see Figure 1).

5.1. I recommend providing more details on statistics significance, differences and tests within the result figures and result (by using legends or captions).    

Response: Legends and different letters to show statistical significance have been added in all figures and table3.

The correction has been made (please see the Figures1, 2, 3, 4 and the table 3).

5.2. Many comments provided in the results should be removed or placed in the discussion. For example: lines 259-262; lines 295-297.  

Response:   The sentences originally found in lines 259-262 and lines 295-297 have been removed.

6) Concerning beans, other articles on this research topic have provided insights into biochemical parameters in addition to growth parameters. Within this context, I recommend that the authors provide perspectives on biochemical and molecular parameters for future studies in the discussion.

Response:  Thank you for your valuable suggestion. We appreciate your input on enhancing the organization of our study. In response to your feedback, we have incorporated the perspectives on biochemical and molecular parameters for future studies in the discussion section in the revised manuscript.

 (Please see p. 14 line 437)

  • Please refer to the following articles (these are just a few examples; there are many others):    https://www.mdpi.com/2073-4395/10/2/189  

                            https://www.mdpi.com/2076-3417/12/2/776  

 “How Does the Addition of Biostimulants Affect the Growth, Yield, and Quality Parameters of the Snap Bean (Phaseolus vulgaris L.)? How Is This Reflected in Its Nutritional Value?”    By the way, I recommend that the authors read and cite these articles to better clarify the novelty of the current research in the introduction and discussion and to provide readers with a better overview of current research in this topic.     

Response:   Thank you for the suggestion. We have understood that you would like us to include sentences concerning the enhancement of the nutritional value of snap beans by biostimulants. Based on our understanding, we have added these sentences in the revised manuscript. We hope that we have made the revisions correctly.

These articles have been referred in introduction and discussion. (Please see p.2 line 53; p. 13 line 388).

6.1) The authors mentioned molecular advances (line 67). Have they provided any molecular insight? If yes, it should be incorporated into the current manuscript.

Response:  the mentioned article holds significant importance in comprehending the relationship between arbuscular mycorrhizal fungi (AMF) and legumes in metal-polluted areas, as it contributes to the identification of specific endosymbionts. Although we did not elaborate on their specific results as they are not directly related to our studies, the article remains crucial for future research in this field.  But in our discussion, we provided examples of studies that highlight the primary effects of fungal symbiosis on soil phosphatase activities. Additionally, these studies demonstrate how it enhances phosphorus absorption by upregulating phosphate transporter genes. (Please see p. 13 lines 415-417).

Dear reviewer, we apologize again for any difficulty you encountered in reading the article. I hope these answers your question.

Round 2

Reviewer 1 Report

The authors addressed all my comments, many thanks for their contribution

Minor editing of English language required

Author Response

Comments and Suggestions for Authors: The authors addressed all my comments, many thanks for their contribution.

  • We appreciate your input on enhancing the organization of our study.

Comments on the Quality of English Language: Minor editing of English language required.

Response:

  • Dear reviewer, we have taken note of your observation regarding the minor editing of English language of our manuscript. We apologize again for any difficulty you encountered in reading it. According to our understanding we have made corrections, especially in the results section, in the revised manuscript. We hope we revised correctly.

Reviewer 3 Report

In the response letter, it is mentioned: “5.1. I recommend providing more details on statistics significance, differences and tests within the result figures and result (by using legends or captions).    

Response: Legends and different letters to show statistical significance have been added in all figures and table3.

The correction has been made (please see the Figures1, 2, 3, 4 and the table 3).”

However, I do not see yet any explanation about the statistics test that have been used in these figures and tables and the explanation on the different letters;

“x. Please revise the entire manuscript accordingly. 

We have diligently made corrections to the article and have taken into account your valuable feedback.”

I hope the authors have really reviewed the entire manuscript according to English usage and writing (as recommended). If not, I recommend that they do it to provide good quality writing to the readers. The authors are responsible for ensuring good quality of English and writing.

Author Response

Reviewer 3

Response: Legends and different letters to show statistical significance have been added in all figures and table3.

The correction has been made (please see the Figures1, 2, 3, 4 and the table 3).”

However, I do not see yet any explanation about the statistics test that have been used in these figures and tables and the explanation on the different letters.

  • We have added explanations about the statistical tests used in the figures and tables, as well as the significance of different letters. (Please see Figures1,3, 4, 5 and the tables 2 and 3).

“x. Please revise the entire manuscript accordingly. 

We have diligently made corrections to the article and have taken into account your valuable feedback.”

I hope the authors have really reviewed the entire manuscript according to English usage and writing (as recommended). If not, I recommend that they do it to provide good quality writing to the readers. The authors are responsible for ensuring good quality of English and writing.

  • Dear reviewer, we sincerely apologize for any inconvenience it may have caused in terms of readability. Taking your comments into consideration, we have made corrections, particularly in the results section, in the revised manuscript. We hope we revised correctly.